# Impaired Hippocampal Long-Term Potentiation and Memory Deficits upon Haploinsufficiency of MDGA1 Can Be Rescued by Acute Administration of D-Cycloserine

**DOI:** 10.3390/ijms25179674

**Published:** 2024-09-06

**Authors:** Daiki Ojima, Yoko Tominaga, Takashi Kubota, Atsushi Tada, Hiroo Takahashi, Yasushi Kishimoto, Takashi Tominaga, Tohru Yamamoto

**Affiliations:** 1Department of Molecular Neurobiology, Faculty of Medicine, Kagawa University, Miki-cho 761-0793, Kagawa, Japantada.atsushi@kagawa-u.ac.jp (A.T.); takahashi.hiroo@kagawa-u.ac.jp (H.T.); 2Institute of Neuroscience, Tokushima Bunri University, Sanuki 769-2193, Kagawa, Japan; 3Department of Neurobiophysics, Kagawa School of Pharmaceutical Sciences, Tokushima Bunri University, Sanuki 769-2193, Kagawa, Japan; tkubota@kph.bunri-u.ac.jp (T.K.); y-kishimoto@pharm.teikyo-u.ac.jp (Y.K.); 4Laboratory of Physical Chemistry, Faculty of Pharmaceutical Sciences, Teikyo University, Itabashi-ku 173-8605, Tokyo, Japan; 5Kagawa School of Pharmaceutical Sciences, Tokushima Bunri University, Sanuki 769-2193, Kagawa, Japan

**Keywords:** E/I balance, MDGA1, D-cycloserine

## Abstract

The maintenance of proper brain function relies heavily on the balance of excitatory and inhibitory neural circuits, governed in part by synaptic adhesion molecules. Among these, MDGA1 (MAM domain-containing glycosylphosphatidylinositol anchor 1) acts as a suppressor of synapse formation by interfering with Neuroligin-mediated interactions, crucial for maintaining the excitatory–inhibitory (E/I) balance. *Mdga1*^−/−^ mice exhibit selectively enhanced inhibitory synapse formation in their hippocampal pyramidal neurons, leading to impaired hippocampal long-term potentiation (LTP) and hippocampus-dependent learning and memory function; however, it has not been fully investigated yet if the reduction in MDGA1 protein levels would alter brain function. Here, we examined the behavioral and synaptic consequences of reduced MDGA1 protein levels in *Mdga1^+/−^* mice. As observed in *Mdga1*^−/−^ mice, *Mdga1^+/−^* mice exhibited significant deficits in hippocampus-dependent learning and memory tasks, such as the Morris water maze and contextual fear-conditioning tests, along with a significant deficit in the long-term potentiation (LTP) in hippocampal Schaffer collateral CA1 synapses. The acute administration of D-cycloserine, a co-agonist of NMDAR (N-methyl-d-aspartate receptor), significantly ameliorated memory impairments and restored LTP deficits specifically in *Mdga1^+/−^* mice, while having no such effect on *Mdga1*^−/−^ mice. These results highlight the critical role of MDGA1 in regulating inhibitory synapse formation and maintaining the E/I balance for proper cognitive function. These findings may also suggest potential therapeutic strategies targeting the E/I imbalance to alleviate cognitive deficits associated with neuropsychiatric disorders.

## 1. Introduction

The functionality of the brain is profoundly reliant on the activity of neural circuits, which are composed of a combination of excitatory and inhibitory neurons. A significant structural component that governs this activity is the excitatory and inhibitory synapses. The dynamics of their establishment are controlled by the orchestrated assembly of synaptic adhesion molecules [1,2]. Among these molecules, the direct interaction between presynaptic Neurexins and postsynaptic Neuroligins and the functional role of this interaction have been the subject of extensive research. Accumulating evidence provided by a wide variety of in vitro and in vivo studies indicates that these molecules play a pivotal role in promoting synapse formation [3,4,5,6,7]. Their significance in human brain function is further substantiated by gene linkage studies, which show a significant association of Neurexin and Neuroligin genes with neuropsychiatric disorders such as autism spectrum disorder, bipolar disorder, and schizophrenia [8,9,10,11]. Neurexins and Neuroligins, along with other molecules that positively regulate synapse formation, termed synapse promoters, have been identified and extensively investigated; however, relatively little effort has been directed towards identifying molecules involved in mechanisms that suppress synapse formation, termed synapse suppressors.

The MAM domain-containing glycosylphosphatidylinositol (GPI) anchor proteins, MDGA1 and MDGA2, represent a recently recognized class of synapse suppressors [12,13]. MDGAs are vertebrate immunoglobulin superfamily molecules tethered to the membrane through a GPI anchor preferably expressed in the nervous system [14,15,16,17,18,19]. MDGAs directly associate with Neuroligins in *cis* to interfere with the interaction of Neurexins with Neuroligins in a concentration-dependent manner [20,21,22]. MDGAs possess the ability to bind to essentially all Neuroligins to varying extents [23,24,25], suggesting that the functional outcome of MDGAs in the regulation of synapse development would be determined by the availabilities of MDGAs, Neuroligins, and Neurexins, including their expressions, subcellular local concentrations, and post-translational modifications. Interestingly, despite the in vitro nature of MDGAs’ association with Neuroligins, MDGA1 has exhibited selective binding to Neuroligin2 to interfere with inhibitory synapse formation without affecting excitatory synapse formation in cell culture studies [21,22]. Furthermore, in agreement with the cell culture study, MDGA1-deficient mice exhibit selectively enhanced peri-somitic inhibitory synapse formation in their hippocampal pyramidal neurons, leading to impaired hippocampal long-term potentiation (LTP) and hippocampus-dependent learning and memory function [26]. These observations indicate that MDGA1 plays a crucial role in regulating inhibitory synapse formation to properly maintain the excitatory–inhibitory balance (E/I balance) in vivo, and further suggest that MDGAs’ expression would affect the regulation of synapse formation and the resultant E/I balance. We previously reported that *Mdga1^+/−^* mice exhibited learning and memory deficits in a certain experimental paradigm [27]; however, it has not been fully investigated yet if the reduction in MDGA1 protein levels would indeed alter brain function.

We therefore analyzed hippocampus-dependent learning and memory function and the hippocampal LTP of *Mdga1^+/−^* mice, of which MDGA1 protein expression is reduced. We report here that *Mdga1^+/−^* mice also exhibited learning and memory deficits with impaired hippocampal LTP. Furthermore, we found that the impaired function observed in *Mdga1^+/−^* mice can be rescued by the acute administration of D-cycloserine, a co-agonist of NMDAR (N-methyl-d-aspartate receptor), which can help to correct a reduced E/I ratio by enhancing functional NMDAR activity. Our findings verified the functional relevance of MDGA1 synaptic suppressors on proper brain function and provided evidence supporting the idea that certain types of neurodevelopmental disorders might be ameliorated by medication in adulthood.

## 2. Results

### 2.1. Mdga1^+/−^ Mice Exhibit Learning and Memory Deficits

We previously reported that *Mdga2* haploinsufficiency enhanced excitatory synapse formation, leading to memory and social deficits [20]. This suggests that the regulation of MDGA2 protein levels is crucial for maintaining proper E/I balance. Additionally, we demonstrated that the loss of the MDGA1 protein in *Mdga1^−/−^* mice enhances inhibitory synapse formation, resulting in memory and sensorimotor gating deficits [26,28]. However, it was not fully investigated whether a reduction in MDGA1 protein levels would lead to behavioral abnormalities.

In *Mdga1^+/−^* mice, the MDGA1 protein in the brain is reduced to approximately half that of wild-type mice [19]. We therefore examined the behavioral abnormalities in *Mdga1^+/−^* mice, comparing them with *Mdga1^−/−^* and wild-type mice. In the Morris water maze (MWM) test, *Mdga1^+/−^* mice exhibited moderate difficulty in reaching the hidden platform (Figure 1A). In the contextual fear-conditioning (CFC) test, *Mdga1^+/−^* mice also showed deficits in memorizing traumatic experiences in the test chamber, similar to *Mdga1^−/−^* mice (Figure 1B). These observations indicate that *Mdga1^+/−^* mice suffer from significantly impaired hippocampus-dependent learning and memory. The complete absence of the MDGA1 protein is not necessary to alter memory function; rather, a reduction in MDGA1 expression is sufficient to cause learning and memory deficits in these mice.

### 2.2. Mdga1^+/−^ Mice Exhibit Deficit in LTP

We next investigated whether *Mdga1^+/−^* mice also exhibit alterations in synaptic plasticity. We prepared acute brain slices containing the hippocampus from *Mdga1^+/−^*, wild-type, and *Mdga1^−/−^* mice, and tested LTP (1 × 100 Hz, 1 s) in their Schaffer collateral CA1 synapses. A significant deficit in LTP was observed in the brain slices of *Mdga1^+/−^* mice, similar to that observed in the brain slices of *Mdga1^−/−^* mice (Figure 2). These findings collectively suggest that the chronic reduction in MDGA1 protein levels impairs LTP, leading to hippocampus-dependent learning and memory deficit. This implies that the regulation of MDGA1 protein levels is crucial for maintaining proper E/I balance, akin to MDGA2.

To further validate the observed LTP deficits across a broader span of the circuit, we optically recorded membrane potential changes in the hippocampal slices using synthetic voltage-sensitive dye (VSD) [29,30]. It has been reported that LTP at distal sites is more significant than that at proximal sites, which may reflect that distal cells receive more inhibition than proximal cells [30]. The evoked membrane potential changes in the CA1 region were recorded using the VSD recording system (Figure 3). We observed significantly reduced LTP throughout the CA1 region, including the distal areas, in *Mdga1^+/−^* mice slices, similar to what was observed in *Mdga1^−/−^* mice slices (Figure 4). Additionally, the reduction in LTP in *Mdga^−/−^* and *Mdga1^+/−^* mice brain slices was more pronounced in the proximal to middle side of CA1 (Figure 4). These observations collectively support the notion that the amount of synapse suppressors, such as MDGA1, plays a crucial role in maintaining proper neural circuits, possibly through the regulation of inhibitory synapse formation.

### 2.3. Acute Administration of D-Cycloserine Ameliorates Memory Deficits in Mdga1^+/−^ Mice

Given that the reduction in MDGA1 protein levels in *Mdga1*-deficient mice mimics a moderate developmental impairment in the regulation of inhibitory synapse formation, which alters the E/I balance sufficiently to cause memory deficits with impaired LTP, it was hypothesized that pharmacological intervention aimed at correcting the altered E/I balance might ameliorate their impaired memory function. This approach is analogous to medication for certain psychiatric disorders caused by the improper regulation of E/I balance. To test this hypothesis, we administered D-cycloserine to *Mdga1*-deficient mice. D-cycloserine, a co-agonist of NMDAR, increases NMDAR conductivity [31], which may help correcting the reduced E/I ratio by enhancing functional NMDAR activity. We found that acute intraperitoneal injection of 10 mg/kg D-cycloserine to *Mdga1^+/−^* mice (1 h before each trial) significantly reduced the time required for *Mdga1^+/−^* mice to reach the hidden platform in the MWM test, while the same treatment did not affect the behavior of wild-type or *Mdga1^−/−^* mice in the test (Figure 5). D-cycloserine administration gave no significant difference in visible platform task (Appendix A). We further examined the effect of D-cycloserine on *Mdga1*-deficient mice using the CFC paradigm. As shown in Figure 6, control intraperitoneal injection of PBS did not alter the percentage of freezing time in the test chamber for either wild-type or *Mdga1*-deficient mice; both *Mdga1^+/−^* and *Mdga1^−/−^* mice exhibited significantly less freezing behavior compared to wild-type mice. However, after intraperitoneal injection of 10 mg/kg D-cycloserine, *Mdga1^+/−^* mice displayed freezing behavior comparable to wild-type mice, while *Mdga1^−/−^* mice continued to exhibit significantly less freezing behavior. These observations collectively indicate that pharmacological correction of the E/I imbalance induced by moderately reduced MDGA1 protein levels can ameliorate deficits in spatial and contextual memory, whereas deficits caused by the complete absence of the MDGA1 protein cannot be rescued by the same treatment.

### 2.4. D-Cycloserine Restored LTP Deficit in Mdga1^+/−^ Mice Brain Slices

D-cycloserine ameliorated hippocampus-dependent spatial and contextual memory deficits in *Mdga1^+/−^* mice. To verify whether the hippocampal LTP of *Mdga1*-deficient mice was altered by the administration of D-cycloserine, we examined LTP in the Schaffer collateral CA1 synapses in the presence of 10 µM of D-cycloserine. We found that D-cycloserine ameliorated LTP in brain slices from *Mdga1^+/−^* mice, making the difference from wild-type mice insignificant, while LTP in brain slices from *Mdga1^−/−^* mice remained significantly impaired in the early phase of LTP induction (Figure 7). To confirm that D-cycloserine actually altered LTP, we compared the field excitatory postsynaptic potential (fEPSP) values of wild-type, *Mdga1^+/−^*, and *Mdga1^−/−^* mice with and without D-cycloserine at 60–80 min post-stimulation. In *Mdga1^+/−^* mice brain slices, the fEPSP values were significantly upregulated, while no significant changes were observed in wild-type and *Mdga1^−/−^* mice brain slices (Appendix A).

To further validate the restoration of LTP defects by D-cycloserine in the broader CA1 area, we optically recorded membrane potential changes in the CA1 region of wild-type, *Mdga1^+/−^*, and *Mdga1^−/−^* mice in the presence of D-cycloserine using VSD (Figure 8). Treatment with D-cycloserine generally improved the induction of LTP in both *Mdga1^+/−^* and *Mdga1^−/−^* mice. Notably, even in the central part of CA1, where the most pronounced impairment was observed, LTP was achieved at levels nearly comparable to those in wild-type mice. However, as observed in fEPSP, brain slices from *Mdga1^+/−^* mice showed no significant differences from wild-type mice over the entire period, whereas *Mdga1^−/−^* mice exhibited significant impairment in the early phase of LTP induction following HF stimulation in the central CA1 region. These observations collectively indicate that the memory deficits caused by *Mdga1* haploinsufficiency were rescued by the restoration of impaired LTP through the acute administration of D-cycloserine. This further suggests that appropriate medication could ameliorate causal deficits even after chronic impairments resulting from a lifelong E/I imbalance.

## 3. Discussion

In this study, we demonstrated that *Mdga1^+/−^* mice exhibit impaired hippocampal LTP and deficits in memory function similar to those observed in *Mdga1^−/−^* mice. These findings underscore the functional relevance of MDGA1, a synaptic suppressor, in maintaining proper neural circuits. MDGA1 is selectively expressed by pyramidal excitatory neurons in the CA1 region [26], suggesting that even moderate alterations in inhibitory synapse formation on these neurons can significantly affect the establishment of LTP. Recent characterizations of MDGA proteins have shown that endogenous MDGAs homogeneously distributed on the neural cell membrane prevent extra-synaptic Neuroligins from prematurely associating with Neurexins [32]. These models suggest that the amount of MDGA protein is crucial for determining the availability of Neuroligins during synaptogenesis. Our observations support this idea; the chronic reduction in MDGA1 in mice altered neural functions, such as hippocampal LTP, to a degree sufficient to impair learning and memory. Along with the fact that *Mdga2^+/−^* mice exhibit alterations in the E/I balance, our findings suggest that the amount and/or availability of MDGA family molecules must be tightly regulated to maintain proper neural circuit function. Various hippocampal GABAergic interneurons contribute to proper neural circuit formation [33,34], and it remains to be examined whether overall inhibitory inputs from these neurons are enhanced or whether inputs from a particular subset of neurons are affected in *Mdga1^+/−^* mice. Another finding in our current study is that the observed hippocampal LTP impairment and the resultant learning and memory deficits caused by the chronic reduction in MDGA1 in *Mdga1^+/−^* mice were attenuated by the acute administration of D-cycloserine, a drug that adjusts the E/I imbalance. D-cycloserine increases the conductivity of NMDAR, thereby enhancing excitatory inputs expected to counteract excessive inhibitory inputs. These observations provided evidence supporting the idea that an appropriate pharmacological intervention correcting E/I imbalance could be effective in ameliorating some symptoms that persist from birth to adulthood. Future studies are needed to test whether other drugs that enhance excitatory inputs or reduce inhibitory inputs are effective in ameliorating the deficits observed in *Mdga1^+/−^* mice.

Our study demonstrated that treatment with D-cycloserine attenuated LTP and memory impairment in *Mdga1^+/−^* mice but not in *Mdga1^−/−^* mice. We previously reported that *Mdga1^−/−^* mice exhibit impairment of prepulse inhibition (PPI) of the startle response, while *Mdga1^+/−^* mice show no significant difference compared to wild-type mice [28]. To examine whether another deficit observed in *Mdga1^−/−^* mice is resistant to amelioration by D-cycloserine, we tested its effect on PPI impairment in *Mdga1^−/−^* mice. As anticipated, the acute administration of D-cycloserine did not ameliorate impaired PPI in *Mdga1^−/−^* mice (Appendix A). One possible explanation is that the increase in inhibitory inputs in *Mdga1^+/−^* mice was more excessive than could be compensated for by the administrated D-cycloserine. If this is the case, pharmacological intervention targeting inhibitory input might help in correcting deficits observed in *Mdga1^−/−^* mice when combined with D-cycloserine, which needs to be examined in future studies. Another possibility is that the complete loss of MDGA1 at birth has caused qualitatively irreversible changes in the systems controlling LTP and memory functions. It has been reported that the acute reduction of MDGA1 in adult mice affects the inhibitory synapse formation of hippocampal pyramidal neurons differently from what is observed in *Mdga1^−/−^* mice [35]. VSD imaging data revealed significant suppression of optical signals in the central region of the slice at later stages following HFS (Figure 4). This suppression may reflect an excessive inhibitory response mediated by GABAergic synapses, whose formation was likely enhanced due to the chronic absence or reduction of MDGA1. It remains to be examined whether acute reduction, chronic reduction and complete absence of MDGA1 have qualitatively different effects on the formation and/or maintenance of neural networks.

Optical recording of membrane potential changes in the hippocampal slices using VSD revealed regional differences in the effect of MDGA1 manipulation on HFS-induced LTP and its recovery with D-cycloserine application. This variation may be attributed to the non-homogeneous interaction of the MDGA1 molecule on HFS-induced LTP. Additionally, the influence of MDGA1 on neural signal processing within the CA1 circuit could contribute to these non-homogenous effects. These aspects are not fully addressed in the present study and warrant further investigation.

In summary, our study demonstrated that a moderate reduction in MDGA1 impairs hippocampal LTP and causes learning and memory deficits, which can be rescued by the acute administration of D-cycloserine. Chronic genetic dysfunction is recognized as a cause of developmental disorders [36], and *Mdga1^+/−^* mice may serve as a model for investigating specific types of neurodevelopmental disorders. Further physiological and pharmacological analyses of *Mdga1^+/−^* mice could contribute to a better understanding of the molecular mechanisms underlying the formation and maintenance of neural circuits.

## 4. Materials and Methods

### 4.1. Animals

*Mdga1*^+/−^ and *Mdga1*^−/−^ mice were generated as described previously [19,26] and housed in environmentally controlled rooms of the animal facility. All experiments were conducted in accordance with state and institutional guidelines. Behavioral analyses were performed on adult (10–16 weeks old) mice, and electrophysiological analyses were performed on P8 mice.

The work described here was carried out in accordance with The Code of Ethics of the World Medical Association (Declaration of Helsinki) for experiments involving humans: https://www.wma.net/policies-post/wma-declaration-of-helsinki-ethical-principles-for-medical-research-involving-human-subjects/ (accessed on 3 September 2024); EC Directive 86/609/EEC for animal experiments: http://ec.europa.eu/environment/chemicals/lab_animals/legislation_en.htm (accessed on 3 September 2024); and Uniform Requirements for manuscripts submitted to Biomedical journals: http://www.icmje.org (accessed on 3 September 2024).

### 4.2. Morris Water Maze (MWM)

For the MWM, the protocols were similar to those previously described [37,38]. Mice were placed in a water maze pool (Eiko Science, Osaka, Japan; 120 cm diameter) containing opaque white water (22 ± 2 °C) with a translucent platform (diameter, 10 cm) that was submerged 1 cm below the surface. Visual cues for orienting consisted of four sheets of paper with black and white geometric designs that were attached to the walls of the experimental room. After pretraining, the hidden platform task was conducted 4/day, at least 1 h apart. During acquisition of the task, the platform location remained static, while entry points were changed semi-randomly between the trials. Mice failing to find the platform within 80 s were manually led to the platform. After each hidden platform trial, mice remained on the platform for 30 s, and they were removed from the platform and returned to their home cage with an escape scoop. On the last day of training, a 2 min probe trial was performed one hour after the last trial with the platform removed. Time spent in the correct quadrant was measured as an assay of spatial memory. Performance was monitored and analyzed with an automated video-tracking system (CleverSys, Inc., Reston, VA, USA). 

### 4.3. Contextual Fear Conditioning (CFC)

A conditioning chamber (320 mm × 270 mm × 270 mm; CleverSys Inc., Reston, VA, USA) was used to induce context + shock associations. The procedure was essentially the same as described previously [20,26]. To facilitate the formation of the contextual representation in the chamber, mice were allowed to explore the chamber for 3 min prior to the onset of the unconditioned stimulus (footshock; 2 s, 0.2 mA). Following footshock, mice were left in the chamber for an additional 30 s before returning to their home cage. As an assay of memory retention, the mice were placed back in the conditioning chamber at 1, 24, or 48 h post-training. Mice were videotaped and their freezing behavior (absence of all movement except respiration) was quantified using the CleverSys FreezeScan system (CleverSys Inc., Reston, VA, USA). The data are presented as the percentage of time freezing.

### 4.4. Electrophysiology

The procedure was essentially the same as described [39]. Hippocampal slices were prepared and transferred to a submerged chamber using the plexiglass ring and continuously perfused with ACSF at a rate of 1 mL/min, heated to 31 °C, and bubbled with 95%/5% O_2_/CO_2_ mixed gas. Glass electrodes filled with ACSF and inserted with Ag/AgCl wire were used as stimulating and recording electrodes to measure the field excitatory postsynaptic potential (fEPSP) in the Schaffer collateral (SC) pathway and the stratum radiatum (SR) of cornu ammonis 1 (CA1). Electrical artifacts were removed from the traces, as shown in the results. A stimulation frequency of 0.033 Hz was maintained throughout the experiment. The stimulation intensity was altered using an electrical stimulator (ESTM-8, Brainvision, Inc., Tokyo, Japan) and the IgorPro (WaveMetrics Inc., Lake Oswego, OR, USA) macro program. Field potential recordings were obtained using a differential amplifier (model 3000; AM Systems, Sequim, WA, USA; low-pass filtered at 3 kHz, high-pass filtered at 0.1 Hz, gain × 100), digitized by analog inputs of ESTM-8 at 10 kHz sampling (an AD converter of 16 bits), and fed into a computer. An analysis of electrophysiological data was conducted for these recordings. 

### 4.5. Voltage-Sensitive Dye (VSD) Imaging

Optical recordings were made simultaneously with electrophysiological recordings. The electrophysiological and optical recordings did not interfere with each other. A brain slice was placed in a recording chamber under epifluorescence optics consisting of two main lenses: an objective lens (f = 20 mm, NA = 0.35; Brainvision Inc., Tokyo, Japan) and a Leica Microsystems projection lens (×1.0), a dichroic mirror (575 nm), an absorption filter (530 nm), and an excitation filter (590 nm) [21,22,29]. The fluorescence was measured and projected onto a CMOS camera (MiCAM02, Brainvision Inc., Tokyo, Japan). The ratio of the fractional change in VSD fluorescence to the initial amount of fluorescence (ΔF/F) was used as the optical signal. The frame rate was 0.6 ms/frame on the MiCAM02 camera (12-bit ADC, 4.5 × 10^5^ well depth, 70 dB).

Some of optical recordings were performed with electrophysiological recordings every 30 s, with each recording session acquiring 512 frames (307.2 ms) after a 200 ms blank exposure to stabilize the light source. The optical signals presented in the following sections were spatially and temporally filtered three times with a Gaussian kernel of 5 × 5 × 3 (horizontal × vertical × temporal directions). An analysis of the optical signals was performed using a procedure developed in the Igor Pro software (ver. 6 for experimental setup control, ver. 8 and ver. 9 for analysis) (WaveMetrics Inc., OR, USA). Details of the optical recording technique can be found in our previous publications [29,30,39,40]. The electrophysiological recordings and the optical recordings were used for the analysis, separately assessed for their quality (i.e., noise due to separate causes), and subjected to subsequent statistical analysis.

### 4.6. Statistical Analysis

Data are expressed as the mean ± SEM (behavioral experiments) or mean ± SD (electrophysiological experiments). Statistical analyses were performed by using two-way analysis of variance (ANOVA) followed by Dunnett post hoc test, one-way ANOVA followed by Dunnett post hoc test, or Student’s *t*-test, as shown in the figure legends. The level of significance of the post hoc tests was set at *p* < 0.05. All analyses were conducted using GraphPad Prism software version 7.0 (GraphPad, La Jolla, CA, USA). A *p*-value < 0.05 was considered statistically significant. 

## Figures and Tables

**Figure 1 ijms-25-09674-f001:**
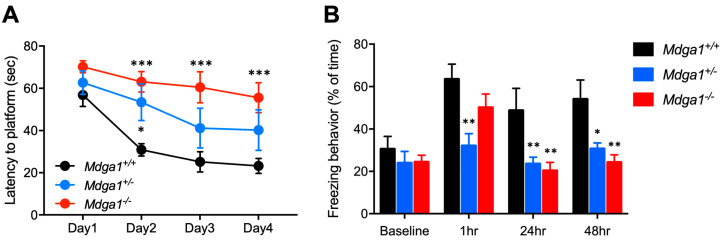
*Mdga1^+/−^* mice exhibit deficits in learning and memory. (**A**) In Morris water maze test, *Mdga1^+/−^* mice exhibit intermediate but still significant deficit in time required for reaching the hidden platform relative to wild-type mice (two-way RM ANOVA, genotype *F*_2,24_ = 9.68, *p* = 0.0008; Dunnett post hoc test * *p* < 0.05, *** *p* < 0.001 versus wild-type mice; *Mdga1^+/+^* mice: *n* = 10; *Mdga1^+/−^* mice: *n* = 7; *Mdga1^−/−^* mice: *n* = 10). Data are expressed as mean ± SEM. (**B**) In contextual fear-conditioning task, *Mdga1^+/−^* mice also exhibit less freezing behavior, as do *Mdga1^−/−^* mice (two-way RM ANOVA, genotype *F*_2,24_ = 6.646, *p* = 0.005; Dunnett post hoc test * *p* < 0.05, ** *p* < 0.01 versus wild-type mice; *Mdga1^+/+^* mice: *n* = 10; *Mdga1^+/−^* mice: *n* = 9; *Mdga1^−/−^* mice: *n* = 8). Data are expressed as mean ± SEM. Analyses of *Mdga1^+/−^* mice were performed along with *Mdga1^+/+^* and *Mdga1^−/−^* mice analyses shown in [26].

**Figure 2 ijms-25-09674-f002:**
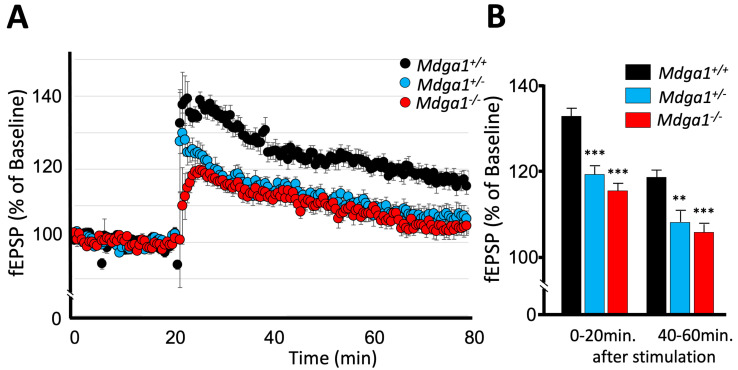
*Mdga1^+/−^* mice exhibit impaired LTP. (**A**) The amplitude of LTP in hippocampal CA1 region following 1 × 100 Hz stimulation. *Mdga1^+/+^* mice: *n* = 12; *Mdga1^+/−^* mice: *n* = 9; *Mdga1^−/−^* mice: *n* = 9. Data are expressed as mean ± SD. (**B**) The LTP of *Mdga1^+/−^* mice slices was impaired at early (0–20 min after stimulation) and later (40–60 min after stimulation) time points, as was that of *Mdga1^−/−^* mice slices. One-way ANOVA followed by Dunnett post hoc test: genotype F_2,27_ = 29.33, *p* < 0.0001 for 0–20 min; genotype F_2,27_ = 13.22, *p* < 0.0001 for 40–60 min. ** *p* < 0.01, *** *p* < 0.001 versus *Mdga1^+/+^* wild-type mice slices. Data are expressed as mean ± SD.

**Figure 3 ijms-25-09674-f003:**
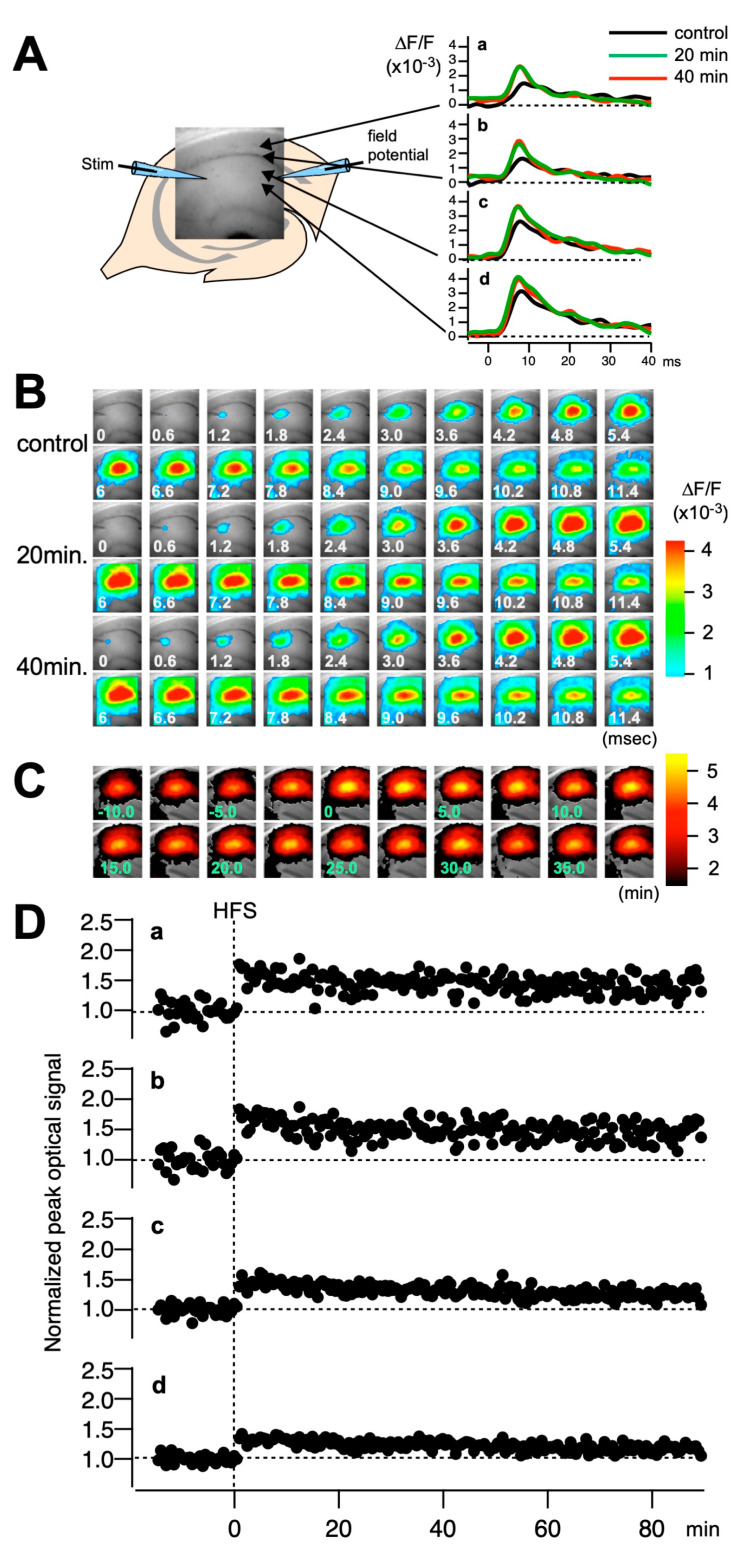
Optical recordings performed simultaneously with field potential recordings. (**A**) A hippocampal slice stained with voltage-sensitive dye (VSD) is shown with a fluorescence image acquired with an optical recording system. Electrical stimulation was applied to the Schaffer collateral with an electrode placed in the stratum radiatum at the CA2/CA1 junction (Stim). The evoked membrane potential changes in the cells in the CA1 region were recorded with the VSD recording system. The rightmost traces are the example of the optical signal recorded at pixels a–d indicated with arrowheads before the application of high-frequency stimulus (HFS) (control, black), 20 min after HFS (green), and 40 min after the HFS (red). (**B**) The consecutive images of the optical signal after the application of electrical stimulation at time zero, taken at the frame rate of 0.6 ms/frame at the control time and 20 min and 40 min after the HFS. (**C**) The maximum amplitude of the optical signal induced by the electrical stimulus mapped on the image (maximum amplitude map) over the series of stimulations at 0.5 Hz. (**D**) The plot of the normalized peak value at the pixels indicated in panel (**A**).

**Figure 4 ijms-25-09674-f004:**
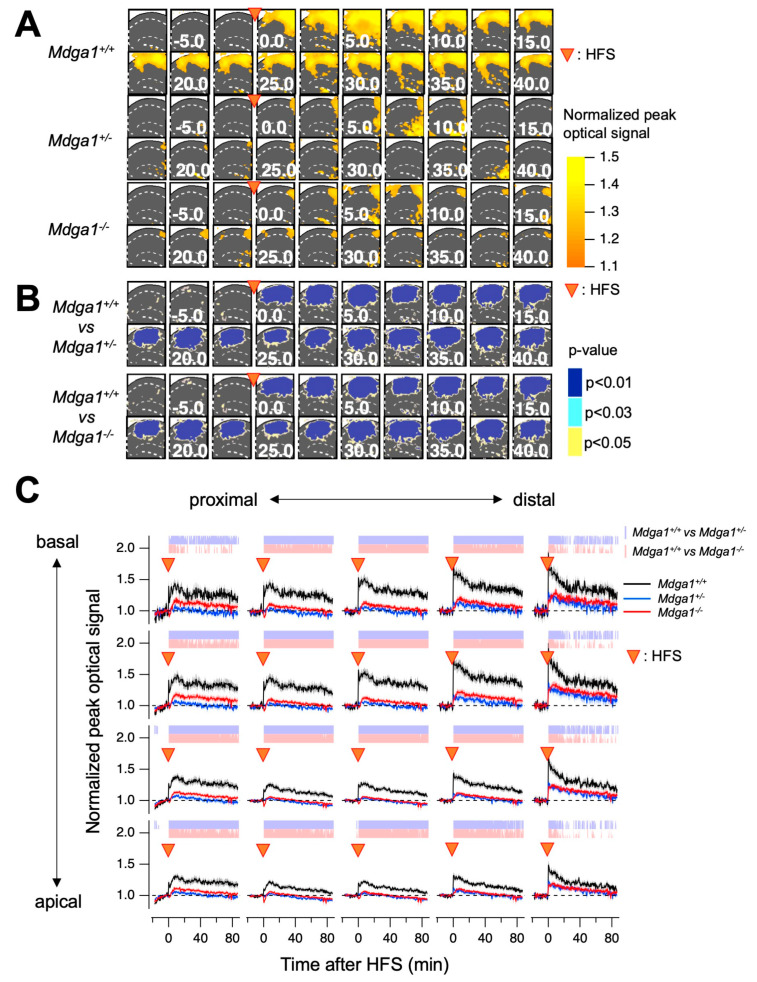
The optically assessed long-term potentiation in *Mdga1* mutant mice. (**A**) The averaged and normalized maximum amplitude maps at different times in *Mdga1^+/+^*, *Mdga1*^+/−^, and *Mdga1*^−/−^ mice. The points at which the HFS was applied are indicated by the arrowheads. (**B**) The *p*-values after statistical comparison on a pixel-by-pixel basis between *Mdga1*^+/+^ and *Mdga1*^+/−^ and *Mdga1*^+/+^ and *Mdga1*^−/−^. The points at which the HFS was applied are indicated by the arrowheads; *p*-values are categorized in the cases of *p* < 0.01, *p* < 0.03, and *p* < 0.05 and colored on the field of view. (**C**) The time course of the normalized amplitude of the optical signal at pixels corresponding to basal and apical over the proximal to distal region of CA1. The points at which the HFS was applied are indicated by the arrowheads. The categorized *p*-values for *Mdga1*^+/+^ and *Mdga1*^+/−^ are shown in blue plots; *Mdga1*^+/+^ and *Mdga1*^−/−^ are expressed in red. The plots are mean ± SEM (*n* = 11 for *Mdga1^+/+^*, 17 for *Mdga1*^−/−^, and 11 for *Mdga1*^+/−^).

**Figure 5 ijms-25-09674-f005:**
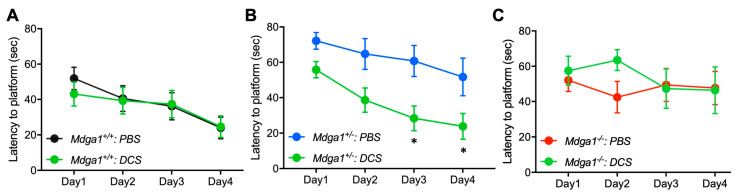
Acute administration of D-cycloserine ameliorated latency of *Mdga1^+/−^* mice for reaching hidden platforms in Morris water maze test. (**A**–**C**) Latency to hidden platforms of mice administered phosphate buffered saline (PBS) or D-cycloserine (DCS) (10 mg/kg; green circles) 30 min before trials. Two-way RM ANOVA followed by Dunnett post hoc test. (**A**) Administration of D-cycloserine gave no significant effect on latent periods of *Mdga1^+/+^* mice. PBS: *n* = 10; D-cycloserine: *n* = 9, medication *F*_1,17_ = 0.0773, *p* = 0.7843. Data are expressed as mean ± SEM. (**B**) Administration of D-cycloserine significantly ameliorated latent periods of *Mdga1^+/−^* mice. PBS: *n* = 8; D-cycloserine: *n* = 9, medication *F*_1,15_ = 7.81, *p* = 0.0136; * *p* < 0.05 versus wild-type mice. Data are expressed as mean ± SEM. (**C**) Administration of D-cycloserine gave no significant effect on latent periods of *Mdga1^−/−^* mice. PBS: *n* = 7; D-cycloserine: *n* = 7, medication *F*_1,17_ = 0.257, *p* = 0.6215. Data are expressed as mean ± SEM.

**Figure 6 ijms-25-09674-f006:**
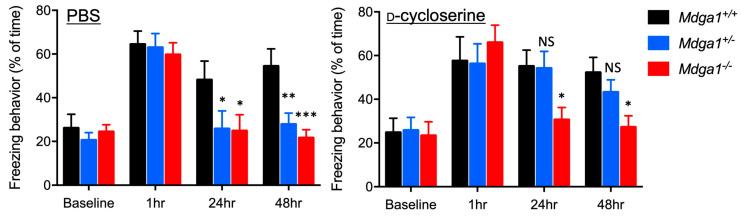
Administration of D-cycloserine restored freezing behavior of *Mdga1^+/−^* mice in contextual fear-conditioning task. (**Left panel**) Freezing behavior of mice administered phosphate buffered saline (PBS) 30 min before trials. *Mdga1^+/−^* mice and *Mdga1^−/−^* mice demonstrated significantly less freezing behavior 24 h and 48 h after experiencing the episode relative to wild-type mice (two-way RM ANOVA, genotype *F*_2,21_ = 7.107, *p* = 0.0044; Dunnett post hoc test * *p* < 0.05, ** *p* < 0.01, *** *p* < 0.001 versus wild-type mice; *Mdga1^+/+^* mice: *n* = 8; *Mdga1^+/−^* mice: *n* = 8; *Mdga1^−/−^* mice: *n* = 6). Data are expressed as mean ± SEM. (**Right panel**) Freezing behavior of mice administered D-cycloserine (10 mg/kg) 30 min before trials. *Mdga1^+/−^* mice demonstrated no significant differences in freezing behavior relative to wild-type mice; however, *Mdga1^−/−^* mice still exhibited significantly less freezing behavior (two-way RM ANOVA, genotype *F*_2,21_ = 1.553, *p* = 0.2349; Dunnett post hoc test * *p* < 0.05 versus wild-type mice; *Mdga1^+/+^* mice: *n* = 8; *Mdga1^+/−^* mice: *n* = 8; *Mdga1^−/−^* mice: *n* = 8). NS: not significant. Data are expressed as mean ± SEM.

**Figure 7 ijms-25-09674-f007:**
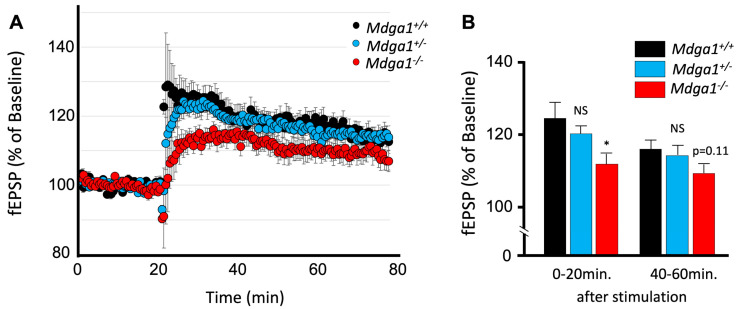
D-cycloserine restored the impaired LTP of *Mdga1^+/−^* mice. (**A**) The amplitude of LTP in the hippocampal CA1 region following 1 × 100 Hz stimulation in the presence of 10 µM D-cycloserine. *Mdga1^+/+^* mice: *n* = 12; *Mdga1^+/−^* mice: *n* = 14; *Mdga1^−/−^* mice: *n* = 10. Data are expressed as mean ± SD. (**B**) Impaired LTP in *Mdga1^+/−^* mice slices, but not in *Mdga1-^−/−^* mice slices, was restored at early and late time points after stimulation. One-way ANOVA followed by Dunnett post hoc test: genotype F_2,33_ = 4.001, *p* = 0.0278 for 0–20 min; genotype F_2,33_ = 2.121, *p* = 0.1359 for 40–60 min; * *p* < 0.05 versus *Mdga1^+/+^* wild-type mice slices. Data are expressed as mean ± SD.

**Figure 8 ijms-25-09674-f008:**
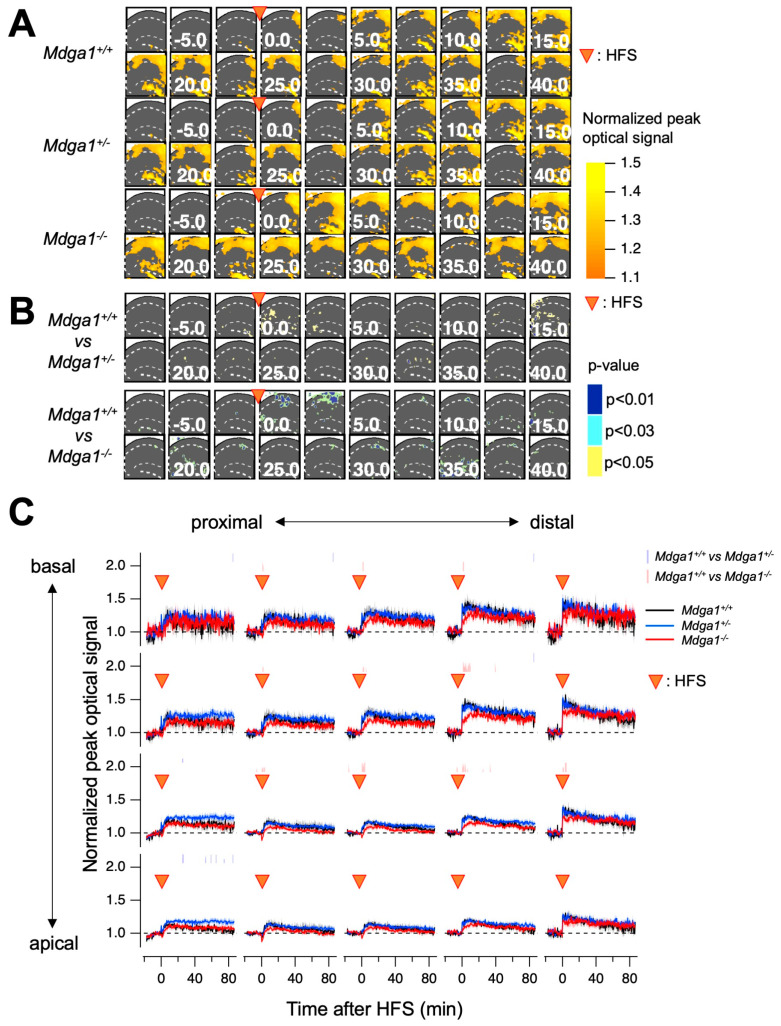
Optically assessed long-term potentiation in Mdga1 mutant mice. (**A**) The averaged and normalized maximum amplitude maps at different times in *Mdga1^+/+^*, *Mdga1*^+/−^, and *Mdga1*^−/−^ mice. The points at which the HFS was applied are indicated by the arrowheads. (**B**) The *p*-values after statistical comparison on a pixel-by-pixel basis between *Mdga1*^+/+^ and *Mdga1*^+/−^ and *Mdga1*^+/+^ and *Mdga1*^−/−^. The points at which the HFS was applied are indicated by the arrowheads. The *p*-values are categorized in the cases of *p* < 0.01, *p* < 0.03, and *p* < 0.05 and colored on the field of view. (**C**) The time course of the normalized amplitude of the optical signal at pixels corresponding to basal and apical over the proximal to distal region of CA1. The points at which the HFS was applied are indicated by the arrowheads. The categorized *p*-values for *Mdga1*^+/+^ and *Mdga1*^+/−^ are shown in blue plots; *Mdga1*^+/+^ and *Mdga1*^−/−^ are expressed in red. The plots are mean ± SEM (*n* = 4 for *Mdga1^+/+^*, 6 for *Mdga1*^−/−^_,_ and 6 for *Mdga1*^+/−^).

## Data Availability

All data generated or analyzed during this study are included in this published article and its Appendix A, except for the VSD data, which is excluded due to its large size but is available upon request from T.T.

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
