# Peer review of "Impaired Hippocampal Long-Term Potentiation and Memory Deficits upon Haploinsufficiency of MDGA1 Can Be Rescued by Acute Administration of D-Cycloserine"

_ijms, 2024, doi:10.3390/ijms25179674_

Round 1

Reviewer 1 Report

Comments and Suggestions for Authors

The paper presents a clear aim and precise results. It examines how the MDGA1 protein impacts learning and memory behavioral tasks and synaptic plasticity by comparing the effects of its absence or reduction. The study demonstrates that reducing MDGA1 protein levels harms hippocampus-dependent learning and memory function and long-term potentiation. These effects are attributed to an impairment in the regulation of inhibitory synapse formation, which leads to a reduced excitation/inhibition (E/I) ratio. The authors also found that administering D-cycloserine, an agonist of glutamatergic NMDA receptor, restored the behavioral responses in learning and memory tasks and the LTP to levels similar to those of wild-type mice, but only for the MDGA1+/- mice and not for the MDGA1-/- mice.

Further elaboration on how MDGA1 affects the E/I balance and potential ways to restore it could benefit the discussion, especially in light of recent studies suggesting no effect of MDGA1 deletion on inhibitory synapse number and transmission.

Author Response

Comments 1:

The paper presents a clear aim and precise results. It examines how the MDGA1 protein impacts learning and memory behavioral tasks and synaptic plasticity by comparing the effects of its absence or reduction. The study demonstrates that reducing MDGA1 protein levels harms hippocampus-dependent learning and memory function and long-term potentiation. These effects are attributed to an impairment in the regulation of inhibitory synapse formation, which leads to a reduced excitation/inhibition (E/I) ratio. The authors also found that administering D-cycloserine, an agonist of glutamatergic NMDA receptor, restored the behavioral responses in learning and memory tasks and the LTP to levels similar to those of wild-type mice, but only for the MDGA1+/- mice and not for the MDGA1-/- mice.

Response 1:

We greatly appreciate the reviewer’s thorough analysis and positive assessment of our study. We are pleased that the clarity of our aim and the precision of our results were well-received. The reviewer’s summary accurately captures the key findings of our work, including the impact of MDGA1 on learning and memory, synaptic plasticity, and the differential effects of D-cycloserine treatment. 

Comments 2:

Further elaboration on how MDGA1 affects the E/I balance and potential ways to restore it could benefit the discussion, especially in light of recent studies suggesting no effect of MDGA1 deletion on inhibitory synapse number and transmission.

Response 2:

We appreciate the reviewer’s insightful comment on the need for further discussion regarding recent observations that acute reduction of MDGA1 in postnatal mice affects inhibitory synapse formation in hippocampal pyramidal neurons differently than in Mdga1-/- mice. In response, we have expanded the discussion on this issue, incorporating our own observations.

Reviewer 2 Report

Comments and Suggestions for Authors

 The article is generally well-structured and clearly conveys the objectives, methods, results, and conclusions of the study. The flow from background information to findings and implications is logical.

 The research topic is timely and relevant, particularly in the context of understanding mechanisms underlying synapse formation, E/I balance, and their role in cognitive functions and neuropsychiatric disorders.

The manuscript clearly presents significant findings, particularly the relationship between MDGA1 protein levels and synaptic and cognitive function, including the effects of D-cycloserine treatment.

 The implications of the findings for therapeutic strategies targeting E/I imbalance are briefly mentioned, which adds translational value to the study.

Overall, the article presents a solid and scientifically relevant investigation into the role of MDGA1 in synapse formation and cognitive function.

Author Response

Comments: 

 The article is generally well-structured and clearly conveys the objectives, methods, results, and conclusions of the study. The flow from background information to findings and implications is logical. The research topic is timely and relevant, particularly in the context of understanding mechanisms underlying synapse formation, E/I balance, and their role in cognitive functions and neuropsychiatric disorders. The manuscript clearly presents significant findings, particularly the relationship between MDGA1 protein levels and synaptic and cognitive function, including the effects of D-cycloserine treatment. The implications of the findings for therapeutic strategies targeting E/I imbalance are briefly mentioned, which adds translational value to the study. Overall, the article presents a solid and scientifically relevant investigation into the role of MDGA1 in synapse formation and cognitive function.

Response:

 We sincerely appreciate the positive feedback and evaluation of our manuscript. We are pleased that the reviewer found the structure, relevance, and scientific contribution of our study to be robust. Based on the reviewer’s comments, no further revisions were deemed necessary. Thank you for your support.